# What Can We Learn from Rural Youth in British Columbia, Canada? Environment and Climate Change—Issues and Solutions

**Pranita Bhushan Udas** [ID], **Bonnie Fournier** *[ID], **Tracy Christianson** [ID] and **Shannon Desbiens**

School of Nursing, Thompson Rivers University, TRU Way, Kamloops, BC V2C 0C8, Canada; pudas@tru.ca (P.B.U.); tchristianson@tru.ca (T.C.); shannondesbiens@hotmail.com (S.D.)
* Correspondence: bofournier@tru.ca

**Abstract:** "What can we learn from rural youth?" was a youth-led arts-based participatory action research project carried out to understand and facilitate positive youth development in two rural communities in the province of British Columbia, Canada. Data was collected using photovoice, visual art, journal reflections, and group discussions. During the study, youth expressed a strong connection with nature for their development or wellbeing. Issues such as environmental degradation and climate change were identified as causes for concern. They discussed human responsibility for environmental stewardship both in their local communities and globally. Climate change hazards such as flood and fire, human action leading to environmental pollution, and human responsibility for environmental stewardship surfaced as issues for their development. Youth expressed a felt responsibility to act on climate change and to reduce the anthropogenic impact on the Earth. Based on youth voices, we conclude that attempts to engage youth in climate action without considering their psychosocial wellbeing, may overburden them.

**Keywords:** rural youth; positive youth development; climate change; climate action; mental health

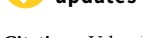



## 1. Introduction

Anthropogenic activities such as the burning of fossil fuels and land manipulations for infrastructure development, commercial agriculture, industrial expansion, etc., have disrupted the natural processes of biotic and abiotic transformation—components of a healthy ecosystem [1–3]. As a result, the Earth has become ill, a scientifically proven fact on global warming and climate change [4]. Increased environmental events such as fire, flood, cyclones, drought, heatwaves and the sudden extinction and emergence of new species are some of the symptoms of the Earth's illness. As noted by the 2009 Lancet Commission, climate change is "the greatest threat to global health of the 21st century" [5]. There are global endeavors to correct the anthropogenic impact on climate. The 2015 Paris Agreement and Sustainable Development Goal 13 on climate action track global commitments to reduce human impact on the climate [6,7]. While the Paris Agreement focuses on policy action, youth engagement in climate action is becoming more of a priority in global development. For example, the Declaration on Children, Youth, and Climate Action signed on 9 December 2019 in Madrid by 10 countries acknowledges the critical roles of children and youth as agents of change. It contains a commitment for action on several youth-related climate goals: the promotion of youth rights, including a right to a healthy environment; investment in youth capacity, including mitigation and adaptation actions; and the promotion of youth participation in climate governance. Indeed, youth aged 15 to 24 make up one-fifth of the world's population and are perceived as a vital force for transformative change, and their engagement is expected to bring positive youth development [8]. Some youth have expressed their concerns regarding the anthropogenic impact on global warming and are already active participants in the global process of negotiations on climate action [9].

Youthhood, is indeed a crucial and vulnerable stage of human development. The transitional nature of youthhood from dependent child to independent adult makes this stage challenging. For many youth, the transition is a determinant to becoming a responsible citizen. For others, this stage is a turning point, where experimentation with substance use may become significant, and where symptoms of psychological distress may begin to surface. For example, suicide is the fourth cause of death of youth aged 15 to 19 in the world and the second leading cause in Canada [10,11]. Considering the fragile nature of youthhood, it is important and even more urgent now to understand youth aspirations and their development in relation to the call for youth engagement in climate action.

"What can we learn from rural youth?" was a youth-led arts-based participatory action research project carried out to understand and facilitate positive youth development in two rural communities in the province of British Columbia, Canada (The research is funded by Social Science and Humanities Research Council of Canada). Youth (aged 14–19 years) were engaged in the research for three years (from 2018 to 2020) through art-based methods to identify issues that mattered to them. Out of the many issues they discussed as opportunities and challenges, issues related to nature and natural disasters is highlighted as the concern. Based on these youth voices, the paper concludes that environmental issues are important elements of youth development, and contemporary issues on climate change are affecting their development. The paper closes by discussing lessons learned regarding youth engagement in climate action.

## 2. Materials and Methods

### 2.1. Context

Canada, the second-largest country in the northern hemisphere, is warming up like many other countries (For further detail see http://prairieclimatecentre.ca/2017/10/seeing-is-believing-historical-records-prove-canada-is-warming, accessed on 18 January 2021). The Canadian government has taken steps to go green and reduce carbon emissions. Global warming and subsequent climate action are a concern for Canada's youth, as reflected in the 2019 Canada Youth Policy, which was prepared in a consultative process with more than 5000 youth who prioritized climate action as one of the six priority areas [12].

Youth make up 19 percent of the Canadian population, of them, one-third are less than 19 years of age and 15 percent of them live in rural areas [11]. Compared to urban youth, rural youth face unique challenges. For instance, rural youth are at higher risk of overall injury as they tend to take risky travel means like motorbikes when public transportation is limited [13]. Mortality rates for most causes of death remain higher in rural and remote regions [14]. Additionally, the possibilities for higher education and employment are more attractive in urban areas and rapid youth migration to cities is a growing phenomenon.

Considering the hurdles for rural youth and their development, two rural areas of British Columbia (BC) located in the boundary of the Interior Health Authority i.e., one of the five health authorities in British Columbia were selected for the study (Figure 1).

Ashcroft and nearby areas of Cache creek and 16 Mile House are in the Thompson Nicola Valley region. Kimberley is in the East Kootenay region. These rural communities were selected, taking into account their geographic and economic variations. Ashcroft and nearby areas are less populated and have relatively lower economic and tourist opportunities than Kimberley (Table 1).

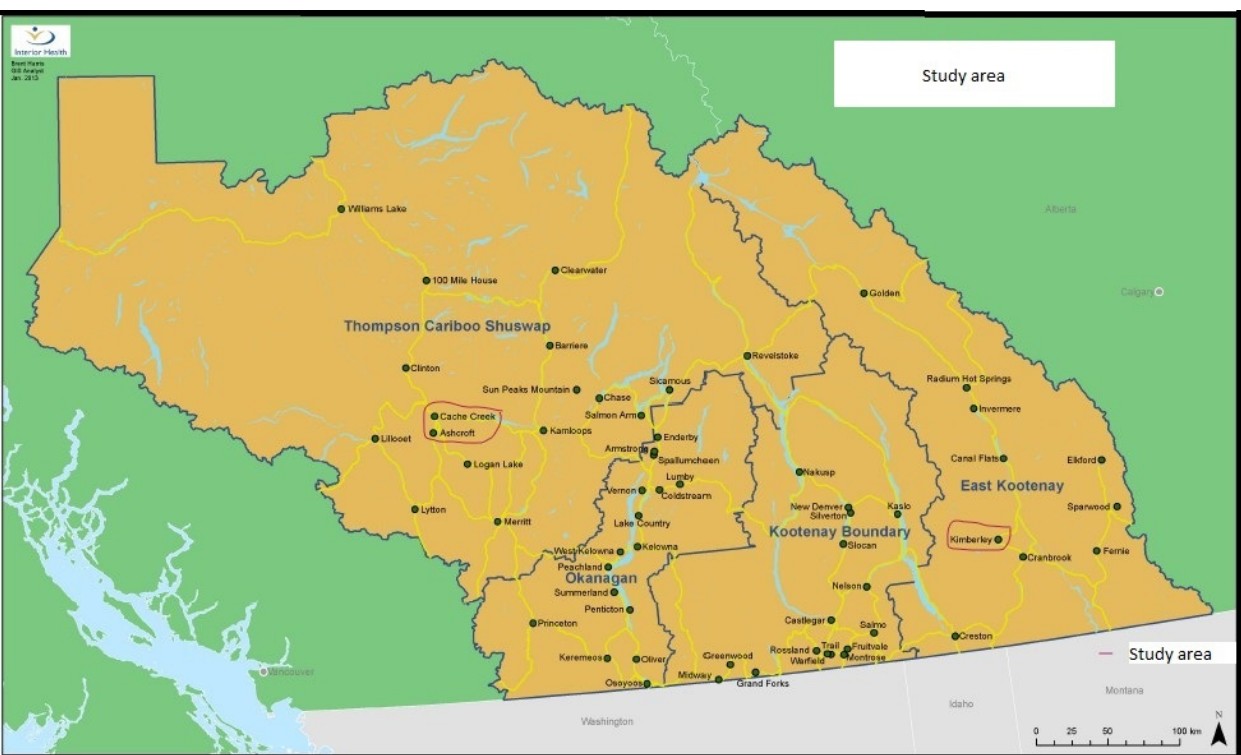

**Figure 1.** Study area within Interior Health Authority, British Columbia. Source: https://www.interiorhealth.ca/AboutUs/Documents/IH_map.pdf (accessed on 6 January 2021).

**Table 1.** The salient feature of study areas.

| Features | Ashcroft And Cache Creek | Kimberley |
|---|---|---|
| Population (2016 Census) | 2050 | 4510 |
| Youth aged 15–19 | 75 | 215 |
| Percentage Youth of total population | 3.4 | 4.7 |
| Population growth trend between 2011–2016 | −6.6% | 11% |
| Administrative Boundary (Regional district) | Thompson Nicola | East Kootenays |
| Elevation | +306 m MASL | +1120 m MASL |

Source: [15–17].

The population growth trend in Ashcroft is declining. The economic opportunity is high in Kimberley, as it actively invites immigrants, and the population growth is positive. Kimberley shares a border with the United States of America. Though both communities were mining villages initially (zinc mining in Ashcroft and lead-zinc mining in Kimberley), with the closure of the mines, employment opportunities have been diversified. Many people in Ashcroft work for a copper mine located 40 km away. However, agriculture has traditionally been an important component of Ashcroft's economy. A significant proportion of land is within the Agricultural Land Reserve. A part of this is accounted for with commercial farmland supplying fruits and vegetables in Western Canada and a cattle company selling organic beef.

For Kimberley, tourism is an important part of the economy. It is one of BC's 14 designated resort municipalities (For more information http://www.whistlercentre.ca/project/1407/, accessed on 27 February 2021) and is well known as a ski resort. Sports such as skiing and hockey invite seasonal visitors every year. In addition, Kimberley receives more than 300 days of sunshine that has been captured to improve the economy through

the promotion of solar power [15]. The Kimberley SunMine project is Canada's largest and the first solar project in BC to sell power to the provincial hydro grid.

The global trend of warming is visible both in Kimberly and Ashcroft. The climate data analysis reveals increasing temperature and precipitation trends [4,18]. For the 1951–1980 period, the annual average temperature was 7.5 °C and 3.7 °C respectively in Ashcroft and Kimberley; for 1981–2010, it was 8.5 °C and 4.2 °C. Under a high emissions scenario, annual average temperatures are projected to be 9.9 °C and 6 °C for the 2021–2050 period, 11.8 °C and 7.9 °C for the 2051–2080 period and 13.3 °C and 9.4 °C for the last 30 years of this century. The summer is becoming drier and fire hazards are increasing. In 2017, Ashcroft and nearby areas experienced historic fire disasters [19], while Kimberley experienced lingering smoke from the province-wide fires.

Similarly, the average annual precipitation for the 1951–1980 period was 270 mm and 602 mm respectively in Ashcroft and Kimberley. Under a high emissions scenario, this is projected to be 3 and 6% higher for the 2021–2050 period, 9 and 11% higher for the 2051–2080 period and in both places 12% higher for the last 30 years of this century.

The longitudinal precipitation data publicly available on World Weather Online indicates a spike or increase of precipitation in 2019/2020, indicating potential flood in the rainy season (Figure 2).

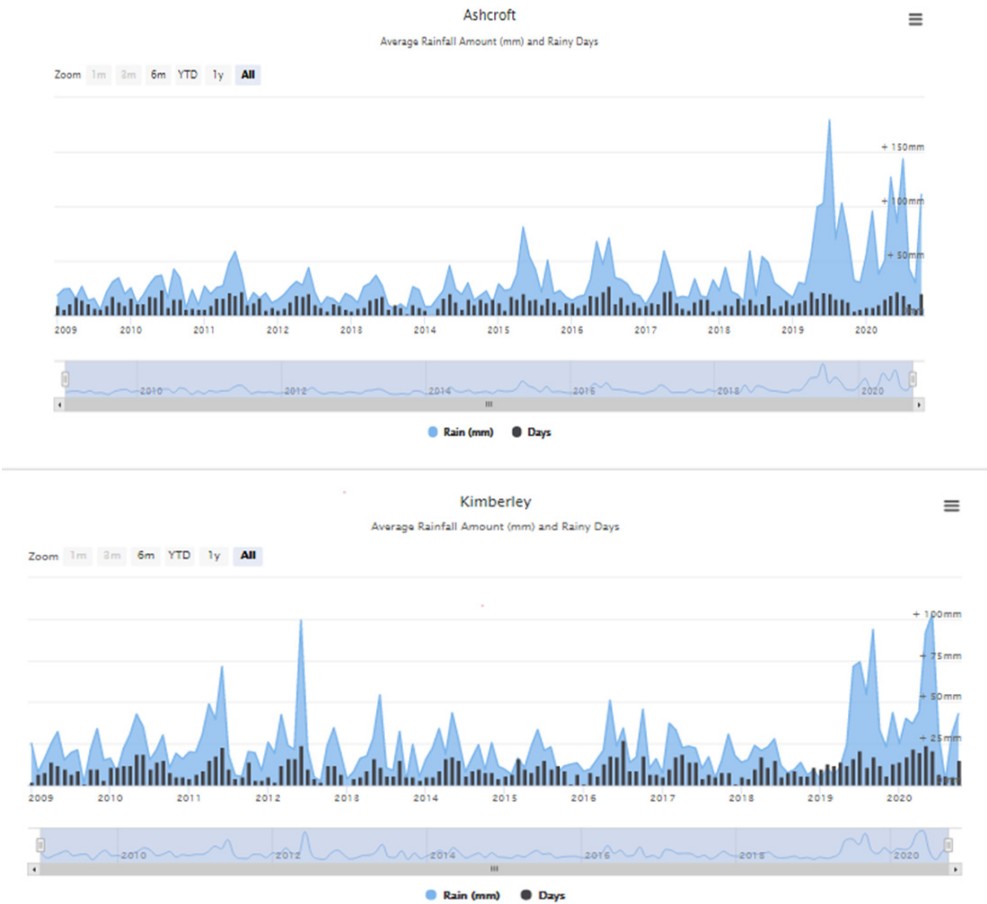

**Figure 2.** Rainfall pattern in Ashcroft and Kimberley 2009–2020. Source: Worldweatheronline.com (accessed on 12 December 2020).

*2.2. Methodology*

For this project, a youth-led arts-based participatory action research approach was used to engage youth in all aspects of the research. The design included four phases of the research process: 1. building relationships, 2. identifying the participants and methods, 3. generating knowledge and planning for change through research, 4. interpretations and

sharing knowledge. Data collection occurred using arts-based methods such as journaling, photovoice, painting, song, and video to facilitate bi-weekly youth discussions. In Ashcroft, three boys and four girls, and in Kimberly, two boys and five girls formed a Youth Action Research Team (YART) in their respective communities. Each YART was youth-led by two selected youth coordinators, who participated in the study for an additional year (2020) to contribute to data analysis and dissemination of the findings. A youth counselor with training on arts-based methods mentored, supported, and facilitated the youth-led discussions.

Nineteen group discussion meetings were organized, ten in Ashcroft and nine in Kimberley. In both communities, the first meeting focused on research training and was followed by three sequential workshops using photovoice. Photos taken by each youth were used as a focal point for discussions. These photovoice discussions were guided by the SHOWeD mnemonic, which is a method of facilitating group discussions in photovoice projects. The SHOWeD questions are: (1) What do you See here? (2) What is really Happening here? (3) How does this relate to Our lives? (4) Why does this situation, concern, or strength exist? (5) What can we Do about it? [20]. The youth took turns discussing what was in their picture, why they took the picture, and what the picture meant to them. After the photovoice sessions, Ashcroft youth decided to focus on music and video activities, whereas in Kimberley, youth opted for painting activities. Youth also engaged in self-reflective journaling on six dimensions of positive youth development, i.e., enhancing connection, competence, character, confidence, caring nature, and contribution for positive change (6Cs) throughout the research process on their engagement with positive youth development (PYD) theory.

### 2.3. A Theoretical Framework: Positive Youth Development and Climate Change

Positive youth development is rooted in enabling youth to thrive [21]. It asserts the belief in youth as resources to be developed rather than problems to be managed [22]. The theory explains five domains of growth or development, i.e., confidence, competence, character, connection, and caring, which, when attained, will lead to the sixth domain contributions by youth [23,24]. These domains are often termed the 6Cs. Literature on PYD emerging from psychology has discussed the relationship between youth development and social environment to promote thriving. Less has been discussed in relation to the impact of the natural environment [25]. Research on ecology, medicine, and adventure studies provides insights on linkages between the natural environment and youth development [26–28]. Hence, to understand youth development in a changing climate context, PYD was combined with theories on the nature-youth connection emerging from multiple disciplines.

Youth engagement with nature that results in positive experiences is found to have a positive impact on their development and nature conservation [26]. For instance, through exposure to nature, youth develop their competence in identifying and addressing risks and challenges [29]. Physical education in school combined with nature adventure has raised awareness among youth about safety, risk, decision-making, problem-solving, and leadership [30]. Physical activities in nature together with family are found to strengthen connection and confidence in youth [31]. However, youth currently face frequent incidences of environmental degradation and disaster, either directly or indirectly, through the media [26,32]. Those who experience a natural disaster are often vulnerable, losing assets and suffering appreciable post-traumatic stress [33]. Other youth who have felt the threat of changing climate are worried about environmental problems and often are pessimistic about the global future [34–36].

While reflecting on the global development discourses, there are variations on hope and threat in different time periods. The narrative in the 1980s was on building new infrastructures such as large dams, irrigation systems, and roads. Other innovations like improving seeds were to address the then problem of hunger and poverty. These developmental goals circulated waves of hope for progress and peace [37,38]. In contrast, the

current discourses are on the global climate crisis and associated uncertainties; moreover, the claim of human actions on carbon emissions and the potential threat of aging infrastructure [39,40]. These discourses provide more worries or threats than hopes and will have implications on the mental health and wellbeing for many.

In addition to the global development discourses, the effect of climate change on the environment, either as a gradual onset, such as changes in temperature, precipitation, and their resulting impact on ecosystems, or as extreme climate events like flood, fire, cyclone, etc., have varied implications on mental health ranging from direct to indirect and short to long term effects [41]. Individuals may suffer from "ecological anxiety" as apprehension or stress about anticipated threats to salient ecosystems and ecological grief in relation to ecological loss [42]. Such ecological anxiety also termed as "climate anxiety" in the climate change context, is more visible among youth [43]. Some people have felt distressed to adapt to a changing environment, termed as solastagia [44]. In addition, climate-induced disaster leaves children and youth with traumatic experiences [45]. Other psychosocial and community effects are in response to climate-induced migrations, climate-related conflicts, and post-disaster adjustments [46].

Though climate change is an integral phenomenon of the Earth's system, stable and gentle changes in climatic pattern are less noticeable [47]. Stable and gentle change provides more certainty, rhythm, and routine on weather patterns to make predictable plans to interact with nature. Such interaction provides positive experiences. In contrast, an unstable natural environment with rapid, unpredictable changes (the current global scenario) increases uncertainties, fear, and anxiety, potentially affecting all six domains of PYD. Moreover, depending on various levels of access to and control over assets or capitals to respond to the climatic threat and loss, youth in different locations or of different gender and socioeconomic categories will have varying capabilities to respond to climate change [48,49]. Youth development in this dynamic nature of a changing climate can be viewed along a continuum, as shown in Figure 3.

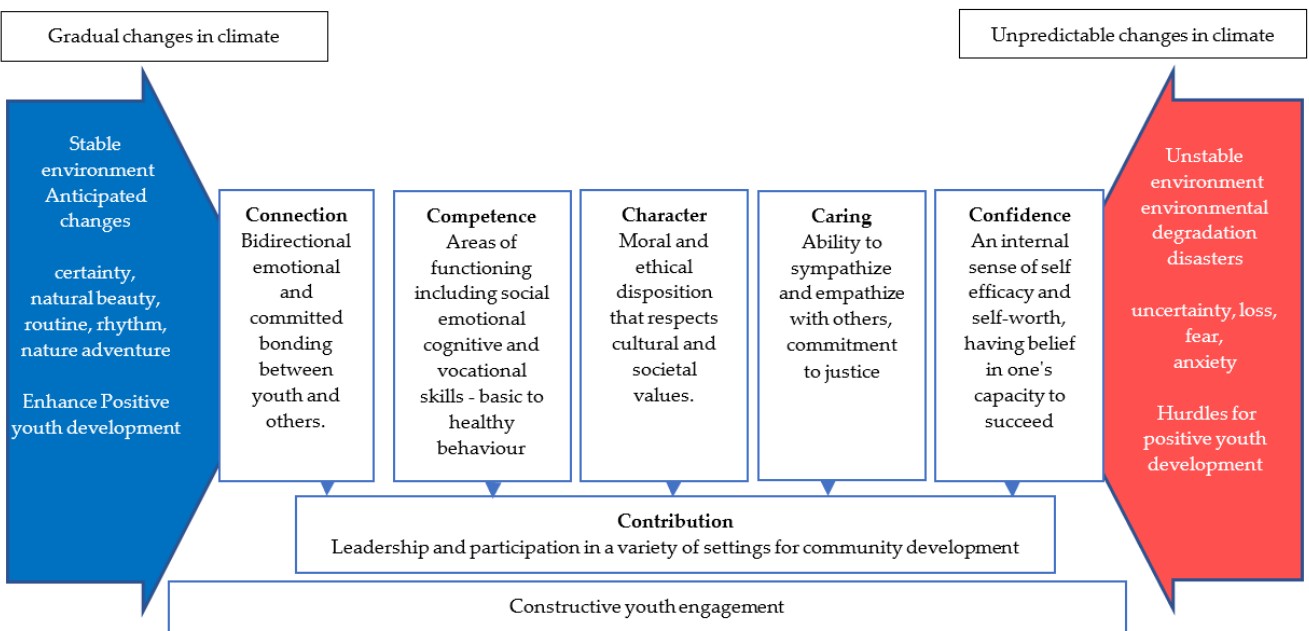

**Figure 3.** Positive youth development within changing climate continuum.

Earlier studies have highlighted the negative impact of climate change on youth and cultivating hope for a better world is emphasized as a way forward [35,36,42]. In this study, we aimed to understand constructive youth engagement as a bridge to support youth for their positive development while transitioning from a gentle pattern of climate change to current rapid and rigorous changes. It is a way forward to achieve positive

youth development as well as to minimize negative human impact on the environment. Engagement with peer-to-peer support and youth mentoring are found to be effective methods of youth engagement and deemed as a protective factor to support youth's sense of belonging and self-efficacy making them less prone to depression and hopelessness [50].

## 3. Results and Discussions

Data analysis occurred throughout the study as an iterative process. Using Miles, Huberman, and Saldana's (2018) work, a thematic analysis was undertaken [51]. Codebooks were created after the final youth discussion meeting. Youth Coordinators reviewed codebooks in a data analysis workshop and created a group art piece to reflect their voice on the key findings. During data analysis, four themes were developed: 1. Youth and connection to the environment; 2. Changing environment and natural disaster; 3. Climate change awareness; and 4. Youth proposed solutions.

### 3.1. Youth and Connection to the Environment

In the first discussions, youth discussed the strength and opportunities, and limitation and challenges of the community. Thirteen of the total fourteen youth expressed nature, especially water bodies and the physical environment as the community strength and opportunities. One youth in Kimberley said, "*We are connected to nature . . . We are one with the trees . . . How many people can say they have a waterfall right in the middle of the town? . . . Oh my god we should have built Kimberley around the waterfalls*" (10 March 2019). Reflecting on the strengths of their community, another youth in Ashcroft stated, "*We have Barnes Lake; people love to go fishing there. We have the slough, fun place to go swimming . . . the river is a nice place to go. The river because it is so fast. It's really dangerous, but it's fun . . . *'. While discussing the river and park in Ashcroft, there was a sense of thrill, adventure, and joy. An Ashcroft youth added, "*The nature park in Cache Creek is a place of joy.*" (4 March 2019).

The feeling of pride is an important component of youth development as it enhances self-esteem, perseverance, and confidence [52–54]. As highlighted above, many of the youths discussed the natural environment as the strengths of their community, including the pride they felt and opportunities available for them to grow and develop to be happy. Youth expressed various natural artifacts that added aesthetic value and provided the opportunity for adventure as a strength. Youth were proud to live in a place where these artifacts are located. Photos were also shared illustrating youth's interaction with nature as an experience of joy, exploration, and growth. "*The mountain is also our friend, hiking and playing in the mountains is endless fun. Our parents don't ever supervise us . . . , . . . we did super dumb things.*" (2 April 2019). Such limitless joy and happiness, enthusiasm, and perseverance shape positive character and confidence [55,56].

The second round of discussions focused on the challenges in the community, which were discussed as hurdles, difficulties, and areas youth would like to see improvement. On this topic, too, youth first expressed issues related to the natural environment. These challenges are related to human impact on the Earth, as well as human responsibility for environmental stewardship, both in their communities and across the globe.

Challenges due to human action included issues on environmental pollution, poor infrastructure maintenance, especially after natural disasters, economic differences to access opportunities for nature adventure, and design of infrastructure to access sunlight. Related to pollution, one youth in Ashcroft pointed out water and air pollution. Another youth talked about litter in the environment as people who do not pick up their dogs' feces. There was also a mention about physical pollution by the train tracks. One said, "*the train leaves rusty pipes, cords and wire. People break glass around there . . . *" (4 March 2019). Another youth in the same community mentioned the remains of a burnt shop after the forest fire in 2017, "*Something to consider for actionable items. We have a burnt shop in our backyard*" (4 March 2019).

Youth in Kimberley mentioned the lack of enough sunlight in one of the school buildings "*There is no natural sunlight anywhere (in the school building) and it is scientifically*

*proven that natural light helps people learn better. Students do not like to go to school. It is like a prison'* (10 March 2019). Challenges emerging from human action in relation to the natural environment is not only physical, but also socioeconomic. Youth expressed that though they like to play in snow, as it provides the experience of adventure and fun, not all youth have equal safe opportunities to enjoy winter adventure activities. In Kimberley, economically poor youth cannot afford to go skiing. They go sledding and play ice hockey instead in a risky public place. One youth mentioned, *"after hockey practice we would go to a place by my brothers and just toboggan down the hill. But then they (the city authority) put the rocks down at the bottom of it and it became super dangerous. Poor guy went down the ice and hit the rocks and could not breathe. But yeah, they put the rocks at the bottom and wrecked the toboggan hill"* (10 March 2019).

The youth in both communities also discussed challenges due to natural processes. This included barriers to accessing opportunities due to natural disasters and the threat of wildlife encroachment. Issues such as floods in Ashcroft and blockage due to snowfall or avalanches in Kimberley are frequent events. One youth stated, *"For past few years, a creek (in Ashcroft) has been getting destroyed because of the spring runoff causing floods . . . climate change . . . "* (10 March 2019). Another youth from Kimberley expressed concerns on how landslides obstruct opportunities, *"I think the challenge for people here is that we don't have access to many things . . . we can't do this because there is landslide* (10 March 2019). *Wildlife in rural communities can also cause concern. "There was a lock down in Kimberley for an hour and half, for a cougar . . . "* (2 April 2019).

From these narratives, we concluded that youth expressed a strong connection with nature. Youth expressed how the natural environments of their communities have influenced their wellbeing. Based on these narratives and results found in the literature, we concluded that the natural environment is an important component of youth development in rural areas [26].

### 3.2. Changing Environment and Natural Disasters

Though youth were strongly connected to their natural environment and shared their joy with nature, their environment is changing. Following the discussion on opportunities and challenges, youth began making photos of their surroundings and discussed various issues in three consecutive discussion meetings. In some of the discussions, youth indicated that the natural environment, which is crucial for youth development, is changing and, in fact, degrading; and will likely have an impact on their development.

In British Columbia, over the past 10 years, there have been approximately 1356 wildfires, and 347,104 hectares have burned over a full fire season [57]. In the summer of 2021, the province experienced the highest temperature 46.6 C ever recorded since 1937, followed by months of wildfire (Further details see https://twitter.com/ECCCWeatherBC, accessed on 28 June 2021). The heatwave and continuous fire for months led the BC government to declare state of emergency till September 2021 [58]. Synchronous fires often made it difficult for fire authorities to control them. The stretch of fire season has extended due to drought in the regions [59]. In Kimberley, youth narrated how they were observing the changing pattern of forest fires, *"I remember when I was a little girl living in Kimberley and in the summer, you never heard of forest fires . . . you didn't even hear about it, you heard about it when Smokey the bear came to the school and was like be careful not to start wildfires. It feels for me the past two years have been crazy different (with fires)"* (14 May 2019). In August 2018, an evacuation order was issued in Kimberley as a fire in the surrounding area expanded quickly, threatening the entire city.

In Ashcroft, a massive forest fire requiring mass evacuation and loss of infrastructure and property took place in 2017. It was followed by a second devastating fire that same year, affecting the whole season. The fire remained for fifty-eight days, and people were evacuated [19]. In Ashcroft, an evacuation became difficult as the notice could not be printed since the electricity and telephone lines were cut off by fire. When electricity was restored, people could not escape since their Bank cards did not work due to the lack

of phone lines and people could not pay for fuel to leave the community. Gloomy skies and lumps of ash falling from the sky remained throughout the summer, both in 2017. Furthermore, Ashcroft was hit by three major flood disasters between 2015 to 2018. In 2015, 2017, and 2018 flooding caused significant damage and death. The reason for flooding in 2015 was a rainstorm, whereas in 2017 and 2018, it was due to snowmelt upstream. The flood in May 2017 killed a firefighter. It was followed by another massive fire in July. Shortly after this disaster, in 2018, a couple died in a mudslide on the Highway in Cache Creek, leaving the community with one trauma after the other. One youth expressed dilemmas and grievances about how to handle such disasters. They witnessed administrative hurdles for the fire department to stop the fire from burning further. One youth recounted their experience, " . . . *all the Ashcroft people came to 16 Mile, but they didn't do anything. They were like this is not in your jurisdiction. So, the fire burned . . .* " (8 April 2019).

There were discussions among the youth in Ashcroft about a particular house that had been damaged by the flood in 2017 and had remained as an eyesore for the community. After the house was damaged by the flood, the owner abandoned the building due to not being able to pay the debt owed. It became a hub for drug users, and later, an incidence of death inside the house occurred, leaving a chilling experience for the community. The youth wondered why this structure had remained so long as an eyesore for the community. They also discussed that the house probably did not have insurance, as insurance companies might have been reluctant to provide their services in flood and fire-prone areas. Youth empathized with victims of natural disaster incidents in their community, "*yeah I heard about how the flood affected them, and how they had to leave their house. I think about your house, and how upset your father is about the fires. It's heart breaking when your home gets destroyed by natural disaster. It could be that there wasn't any insurance or that insurance didn't cover the flood. That is why they had to abandon it, that there is no money for it and that is their predicament*" (8 April 2019). One of the youths reflecting on the abandoned house and its impact on society, referring to sad memories that the house brought up, said, "*I imagine how the person feels in relation to this house, that there is a lot of bad feeling . . .* " (8 April 2019). With respect to changing environments, youth expressed a feeling of sorrow when damage left by the disaster was not cleaned up quickly. This finding supports other research that has found many young people experience sadness, anger, fear, and a sense of powerlessness regarding climate change and its impacts [60–63].

*3.3. Worries and Dilemas—"So, We Scream. We Are Scared . . . "*

Findings from the photovoice activities also offered insight into youth's concerns on climate change and their worries and subsequent feelings of responsibility to do something about the state of the world.

Of the 14 photos in Ashcroft and 12 photos in Kimberley, five were appreciative in nature, such as appreciating the beauty of a place and the success story of the river cleaning up, whereas all other images raised youth concerns about the need for improvements in their community. With this, it can be inferred that youth are living in an environment where they see more community challenges than strengths and opportunities. This may be due to the lingering effects of disasters and residual trauma that is experienced. For instance, with respect to the house that was flooded and subsequently abandoned, one youth mentioned, "*some people were telling me that they (owner of the house) still had to pay for it (the house) and some people were telling me, they just left. I do not know what the deal is, but it is kind of an eye sore. And people were just smashing windows and people would go in there in Halloween and play stupid games and think it is cool, but it is kind of depressing*" (8 April 2019). Youth are found to be more vulnerable than adults in cases of natural disaster. For instance, a meta-analysis of 96 studies found that youth are vulnerable to appreciable post-traumatic stress after a disaster [33]. Other studies show that youth are susceptible to environmental trauma and more likely to become depressed in case of loss and injuries [64,65]. The above youth narrative on the flooded and abandoned house demonstrates how an undealt with loss became an ongoing source of upset for youth and their community.

In addition to the observed impacts of fire and flood, youth were also aware about the current discussion on the anthropogenic impact of climate change and the call to correct human action. These are mostly discussed in school. Media, such as TV shows "B*ill Nye the science guy*" is equally influential. In Kimberley, some youths were active participants of Greta's movement on climate activism called "Fridays for Future" started in 2018. "Fridays for Future" is a global climate strike movement that started in August 2018, when 15-year-old Greta Thunberg began a school strike for climate (for further details: https://fridaysforfuture.org/what-we-do/who-we-are/, accessed on 20 June 2021) It was globally relevant at the time of the study period in 2019.

After three photovoice workshops, Kimberley youth chose oil on canvas paintings to discuss issues of concern. The art pieces they presented at the 6 May 2019 meeting related to their awareness about environmental degradation, climate change, and the need for youth mental health (Figure 4).

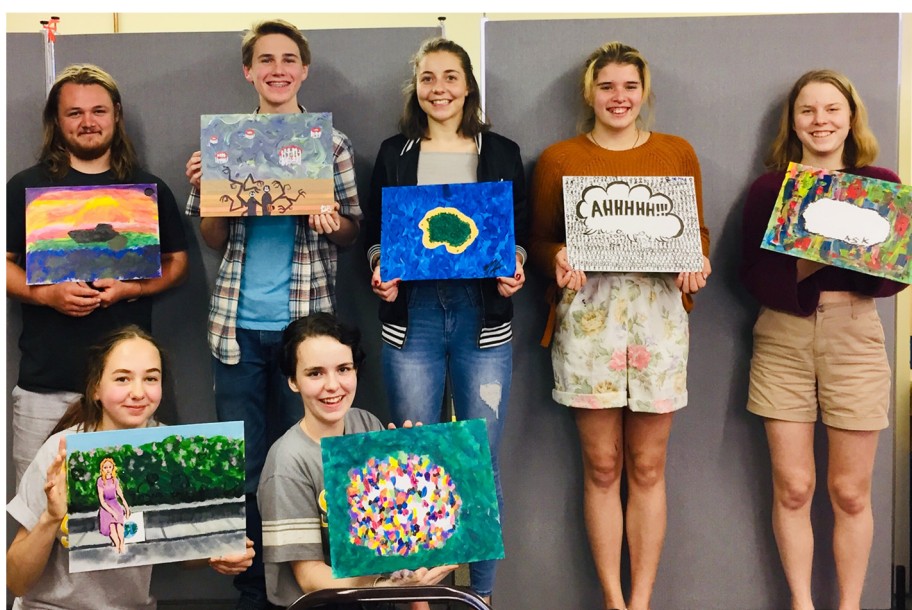

**Figure 4.** Kimberley youth with their artwork.

One youth concerned about pollution in the oceans said, "*My painting represents the Great Pacific Garbage Patch and how it is affecting our oceans, I chose to focus it on climate change as I believe it is such a global issue recently that should be taken more seriously.*" Another youth raised concerns on economic goals of industry ruling over ecological concerns while highlighting ocean pollution, "*Oil and gas are dubbed as "The Villain," but it is a very small portion of the problem. Oil tankers are a risky way of transporting oil across seas; there is no denying that, yet the overlooked high polluter is cruise ships. Cruise ships are just as big, if not bigger, plus they dump sewage and garbage into the ocean. Tourism is not a clean and green profit.*"

Another youth painted Greta highlighting the need to consider youth voice, "*This painting is referenced of Greta Thunberg, the girl that protested climate change by sitting out of school. She ended up getting a lot of support and making global news. I think voices like hers need to be heard more because there are so many more people that have such valuable things to offer the world yet aren't getting a voice. Some groups of people, like youth, aren't getting recognized or being supported, even though they do have a lot that they want to voice. I think we could take the example of Greta's story and take a moment once and a while to give more people a voice.*"

The fourth youth reflected on leisure activities and their effect on the environment, "*My painting began with a mental health theme inspired by the Blue Wave initiative, which represents how mental health has ups and downs. Then I decided that I could not just have a blue painting. I put a tropical island that represents climate change since we all have this idea of the perfect vacation in the Caribbean. However, we do not realize that our perfect islands are threatened*

*because of changing weather patterns, etc. Hurricanes, tsunamis, and ocean waste, in general, are ruining the beautiful places on Earth. As well, these idealistic vacations often include snorkeling or water sports, but the reefs are dying, which is endangering these places even more.*"

Moreover, the fifth youth voice consolidated various polarized or hopeless discussions around them and how it affects their mental health, "*This piece depicts the emotional response youth express when tasked with curing the world. Global warming, abortion laws, sexism, racism, truly the list is endless. So, we scream. We are scared, and we are many, and we are scared.*" These narratives consolidate awareness on climate change issues, youth voice on the need for critical thinking while talking about climate action.

These narratives also partly emerged due to youth observation of the difficulties of '*going green*'. For instance, youth in Kimberley witnessed efforts on climate action that left them with doubt, confusion, and unclear pathways. The SunMine project in Kimberley, the first solar project in BC to sell power, began commercial operation on 22 June 2015 [66]. The people of Kimberley were excited to have this 5.3-million-dollar project as it was envisioned to be self-sustaining and profitable. The performance of the system turned out to be only 90 percent of what was projected and in 2019, the city sold it to a private company due to the financial burden of the running of the system.

The downfall of the SunMine project left doubt for the youth about opportunities and challenges of "going green". The youth said, " . . . *it is a good theory, but not done well . . . Kimberly is in debt that we didn't need to be. Having solar was a good idea. A step in the right direction, hydro power is definitely something we need to start focusing on, because it is not the worst and solar power technology, as we can tell, isn't good enough to get the job done*" (14 May 2019). Moreover, youth challenged the assumption made behind installing solar. The observed hypothesis for this project was that because Kimberley received more than 300 days of sunshine that solar panels would be a good investment. Youth questioned, "*wasn't there a huge problem producing enough energy last summer because of the smoke* (from forest fires)" (14 May 2019).

In another instance, one youth highlighted how the thought about the world dying was stressful. " . . . *if you think about it right from as early as we can remember, the world is dying and you have to do something about it, and we are like we don't know we just have to do this because this is going to happen, . . . what am I going to do, my world is dying and you are like what can I even do? so even when you are not thinking about it, it is another stress in your mind*" (17 April 2019).

These youth narratives showed their awareness and critical thinking on the climate change debate but also the stress and burden that comes with a felt responsibility to heal a "dying world". Acknowledging youth emotions is the first step in developing youth as Earth stewards [67].

*3.4. Youth Proposed Solutions*

Success stories that cultivated hope, critical examination "on going green", and less consumeristic behaviors are some of the solutions youths put forward to tackle the threat of climate change. One such story was discussed by a youth in Kimberly when they shared their photo about environmental pollution, "*This photo of Mark Creek represents what we've done to help the environment, and what can be done to make change in the future. While the Sullivan mine was running, the waste all got dumped into the river, so it ran orange with chemicals. Now, all the chemicals have been cleaned up and people even go swimming in the once toxic water. I think that the story of this needs to be shared with youth because it often feels that environmental issues are piling up and there is nothing*" (17 April 2019, Kimberley). This story is an illustration of correcting mistakes and cultivating hope.

Youth also discussed consumeristic behavior as problematic and the reason for causing environmental degradation with their proposed solution of shopping at thrift stores. "*I think we don't know that consumerism is actually a big problem (but it is). That is why I try to go to thrift shops as much as I can, even though its recycled clothing. And it is not as much as buying a brand-new 300$ hoodie*" (17 April 2019, Kimberley).

Another youth continued that one alone cannot do anything to stop or reverse environmental degradation, whilst emphasizing the need to be connected to nature as a solution to overcome the fear that the Earth is dying, "*That is why I think a lot of people feel so relaxed in nature because you are told your whole life that the earth is dying but when you are in nature, you are like, it's here, I think it is because that stress leaves your mind. But as soon as you go back into the city you are like there are no trees this is what the whole world looks like we are all going to die*" (17 April 2019, Kimberley).

A responding youth expressed the need to have a critical analysis on the reasons for climate change and claimed solutions, "*yeah we should control what we buy as a consumer, but it's like you said, it is a way larger number or statistic, the percentage of carbon emissions that wreck the world are caused by the top 1% of the business world, can't that 1% if they are so rich, shouldn't there be laws to restrict that in any way?*" (17 April 2019, Kimberley). Moreover, youth highlighted the need to ask and involve their voices in the larger discussion on climate action, "*I think climate change is a big thing and a lot of people don't know enough about it . . . and a lot of what they are doing is scaring us about it, like your world is being destroyed and we are just sitting here being like what do I do?*" (17 April 2019, Kimberley).

In Ashcroft, youth discussed the need to improve communication within the community to build social supports and networks as a solution. One youth said, "*The people at 16 Mile are no longer connected. No one communicates with each other anymore. There are not any teens there either, so he is alone in the community. We thought that maybe there could be a block party type of thing where everyone gets together to become more of a community*" (8 April 2019, Ashcroft).

A literature review of studies conducted between 1993 to 2018 argued that most of the solutions proposed by youth in other places are more generic and suggest solutions for which they are not primarily responsible such as planting trees or reducing pollution from factories and transportation [68]. In this study, rural youth proposed solutions that they have been practicing, such as reducing consumeristic behaviors, responsible disposal of refuse, cleaning local sites of pollution or debris, appreciating nature, and valuing environmental stewardship.

## 4. Constructive Youth Engagement—A Way Forward

In Canada, climate action is one of six priorities of Canada's Youth Policy (2018), advocating for increased protection and conservation of the environment. Understanding how to constructively engage youth in climate action is necessary if youth are going to create sustainable change. Key lessons can be drawn from the implementation of a youth-led arts-based participatory action research study to enhance positive development for future work in constructive youth engagement in climate action. Youth-friendly spaces utilizing arts-based research methods provided a place to share and be creative during the research process, which is an empowering experience. One youth coordinator narrated while reflecting on the arts-based research process, "*I believe that the art side of this research is what allowed it to flourish. Getting youth to express their ideas and opinions through various forms of art allowed them to be more comfortable with the group and expand their thoughts. Through the use of arts youth (peers) were . . . able to contribute to the project in unique ways allowing them to grow their 6C's at a more advanced level. Through journaling our 6C's we were able to see not only the change, we were able to create for our community, but [also] the change we created in ourselves*" (20 May 2020).

The opportunity to share among their peers was found to ease youth worries by supporting their positive youth development. As described by another Youth Coordinator, "*A lot of what we talked about wasn't always relevant or pertinent to the themes or action plans, but it resulted in a stronger sense of self and our place in the community*" (20 May 2020). Youth expressed positive energy emerging out of the frequent meetings they had as co-researchers where one stated, "*The students who participated in the project were finally able to feel like they had a voice that mattered. We created an outlet for them to talk about what concerned them and provide guidance to create change*" (20 May 2020). These gatherings allowed peer to

peer support with adult mentorship as needed to express and talk about the issues that mattered to them most. The role of committed and caring adults cannot be overemphasized for promoting thriving youth, whereby youth feel more in control of their lives [56,69]. Research shows that collective engagement on environmental issues is related to hope and wellbeing, perhaps because feelings of efficacy increase when a community is involved, and people can support one another [36]. Within the context of youth climate action, positive youth development may be a protective mechanism to buffer the negative consequences of engaging youth to create change.

## 5. Conclusions

Youth who engaged in this youth-led arts-based participatory action research expressed their concerns on global and local issues. The findings indicated that youth are concerned about climate change and feel responsible for environmental stewardship. They are keenly aware of the harmful impacts of ecological changes driven by the economic and consumeristic goals of societies. While youth feel very strongly about society's role of environmental stewardship, they also expressed a heavy burden to correct the mistakes of past generations they believe are anthropogenic. Based on the findings, we argue that involving youth as an active force in climate action needs careful consideration of their psychosocial development; if not youth may experience undue distress that negatively impacts their positive youth development. Providing opportunities for youth to come together in creative collaboration to support one another in the call to climate action promotes solidarity, resilience, and hope. Moreover, this study highlights the unique challenges faced by rural youth, indicating that their experiences are different from urban youth. Concerns of rural youth need to be addressed, keeping in mind that their life experiences are different and more closely connected to nature and the environment.

## 6. Limitations

While participatory action research can have a transformative effect on those involved because it is inclusive and empowering, as was demonstrated in this study, there were some limitations. The study only compared two rural communities within British Columbia, Canada. To determine if the findings extend beyond these two, other communities across Canada and beyond would provide greater understanding. Comparing diverse socio-cultural diverse communities necessitates further research.

**Author Contributions:** P.B.U. conceptualized the paper, data analysis, writing—original draft preparation; B.F. and T.C. conceptualized the project, data analysis and writing—review and editing, funding acquisition; S.D. coordinated the research, was involved in data collection, writing- review and editing, and was supervised by B.F. and T.C. All authors have read and agreed to the published version of the manuscript.

**Funding:** This research was funded by the Social Science and Humanities Research Council of Canada https://www.sshrc-crsh.gc.ca/home-accueil-eng.aspx, accessed on 2 December 2021.

**Institutional Review Board Statement:** All subjects gave their informed consent for inclusion before they participated in the study. The study was conducted in accordance with the Tri-Council Policy Statement: Ethical Conduct for Research Involving Humans and the protocol was approved by the Ethics Committee of Thompson Rivers University (Project Certificate #101947).

**Informed Consent Statement:** Informed consent was obtained from all subjects involved in the study. Ethics approval was granted by the relevant Human Research Ethics Committee approval number 101947.

**Data Availability Statement:** The data that support the findings of this study are available on request from the corresponding author, B.F. The data are not publicly available due to containing information that could compromise the privacy of research participants.

**Acknowledgments:** We would like to acknowledge the indigenous territories where the study took place Kimberley/Ktunaxa; Ashcroft and Cache Creek/Nlaka'pamux; Kamloops/Tk'emlups: Secwepemc; youth participants in Ashcroft and Kimberley, British Columbia, Canada, nursing practicum students Angela and Taleighla; Brazilian research intern Luana and our collaborators Mark Moody, Ministry of Children and Family Mental Health; Lori Joe, Kootenay Basin Trust Youth Action Network; Candice Estrela, College of the Rockies; Andrea Burrows, Interior Health Authority. Special thanks to the Insight Development Grant of Social Science and Humanities Research Council, Canada, that made this study possible.

**Conflicts of Interest:** The authors declare no conflict of interest.

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
