# Peer review of "What Can We Learn from Rural Youth in British Columbia, Canada? Environment and Climate Change—Issues and Solutions"

_sustainability, doi:10.3390/su132413562_

Round 1

Reviewer 1 Report

This study is particularly valuable as it offers an intriguing way to involve youth in the discussions about the ecological crisis facing our planet that has the potential to enhance their development and encourage their continuing participation in solutions. This contrasts with the more common educational strategies that pummel youth with mountains of information about the overwhelming ecological crisis that we humans have caused and are continuing to cause, that, in turn, can lead to “ecological anxiety” and “ecological grief.” It thus sends an important message: that engaging youth in peer-to-peer interactive educational strategies about ecological issues that are facilitated by highly committed adults can be a protective factor to support youth’s sense of efficacy and so make them less prone to depression and helplessness.

Author Response

Thank you for the positive note on the significance of this paper. Based on your review, we have reviewed the language and spelling; and have improvised accordingly.

Reviewer 2 Report

The article is very interesting and brings the voice of the younger generation who have to deal with climate change. The research tools are varied, although I would have been happy to see the photographs of the participants performed using photovoice. I think it is necessary to edit a few changes before the article can be published.

There is a need to create a more adequate link between the introduction and the discussion by refining the research question. In addition, I think it is necessary to add to the theoretical and practical contribution of the research and what gap it fills.

There is a need to tighten up the results and discussion.

Methodology. From the description of the methodology chapter, I'm not sure if this is a pure Participatory Action Research. Reason and Bradbury (2001, 1) define PAR as “a participatory, democratic process concerned with developing practical knowing in the pursuit of worthwhile human purposes.” According to the authors the students chose only the research tool. I think it's worth sharpening this point.

It is not clear whether steps 3 and 4 in the participatory action research were reflected in the study and the article (line 133-136)

It is not clear what is the age of the study participants

There is a need to address Figure 3 in the discussion, which if not, what is its contribution to the article?

The title of Chapter 3.1 declares a connection to the environment and the opening sentence speaks of the community. It is not clear how things are related to each other

Chapter 3.1, which is supposed to deal with connection to the environment, is not sufficiently focused. For example, it mentions social injustice, but it is not clear whether it affects the connection to the environment. This is one example and there are more. it is necessary to focus the writing in connection to the environment.

Chapter 3.2 contains descriptions that fit the context of the study such as the number of fires and the temperature. This chapter needed rewriting.

It is not clear how the example of the solar project in chapter 3.3 relates to the title of the chapter

A summary of Chapter 3.3 does not indicate what is written in Chapter. While there is example of critical thinking, for example in solar procurement, but the focus is on the financial issue and economic decisions. The examples presented in the chapter show how much children are aware of environmental problems, which is very impressive.

The title of chapter 3.4 is Youth Proposed Solutions but lines 464-470, 476-479 and 480-484 related to the title of the chapter

I think what is missing, throughout the article, is what is the research question of the study?

Author Response

Detailed review and response

  1. The article is very interesting and brings the voice of the younger generation who have to deal with climate change. The research tools are varied, although I would have been happy to see the photographs of the participants performed using photovoice. I think it is necessary to edit a few changes before the article can be published.

Response: A photograph of youth with their art piece has been included in the article to show youth concerns on environmental issues. The comment “I think it is necessary to edit a few changes” is not clear as to what is expected to edit except these comments from Reviewer 2:

  1. There is a need to create a more adequate link between the introduction and the discussion by refining the research question. In addition, I think it is necessary to add to the theoretical and practical contribution of the research and what gap it fills.

Response: Please see the revision on introduction section line 59 to 66, where the research question, discussion and conclusion are linked and highlighted.

  1. There is a need to tighten up the results and discussion.

Response: See revisions in text

  1. From the description of the methodology chapter, I'm not sure if this is a pure Participatory Action Research. Reason and Bradbury (2001, 1) define PAR as “a participatory, democratic process concerned with developing practical knowing in the pursuit of worthwhile human purposes.” According to the authors the students chose only the research tool. I think it's worth sharpening this point.

Response: the Methodology section in line 152 to 156 highlighted the questions that guided the discussion around a picture/photo made by youth. These questions are guiding tool to promote participatory engagement of youth to identify issues that mattered to them. This section also highlights that youth were not only the object of data collection, but also researchers themselves (co-researchers) and were even trained in data analysis and analyzed the transcripts, which makes our study very unique. Youth are not often involved in analyzing data. We have revised the section to make it more clear.  

  1. It is not clear whether steps 3 and 4 in the participatory action research were reflected in the study and the article (line 133-136)

Response: The methodology section highlighted the methods and steps followed in the research. The later section in the results and discussion, highlighted the issues identified in the research following the research process. Since each section has specific objective, the content presented in each section is hence different, and have less scope to repeat the steps highlighted in methodology to present in results and discussion again.

  1. It is not clear what is the age of the study participants

Response: The age of participants (youth) is mentioned in line 57-58 in the Introduction section.

  1. There is a need to address Figure 3 in the discussion, which if not, what is its contribution to the article?

Response: Figure 3 summarise para 205 to 215 of original document. We have revised reference to figure 3 to make it clear.

  1. The title of Chapter 3.1 declares a connection to the environment and the opening sentence speaks of the community. It is not clear how things are related to each other

Response: Title revised to make it precise.

  1. Chapter 3.1, which is supposed to deal with connection to the environment, is not sufficiently focused. For example, it mentions social injustice, but it is not clear whether it affects the connection to the environment. This is one example and there are more. it is necessary to focus the writing in connection to the environment.

Response: Revised – see in text changes

  1. Chapter 3.2 contains descriptions that fit the context of the study such as the number of fires and the temperature. This chapter needed rewriting.

Response: Section 3.2 highlights that the environment in which youth are living is changing. The data on fires incidences are presented to highlight the drought scenarios. It is not clear what is expected to revise.

  1. It is not clear how the example of the solar project in chapter 3.3 relates to the title of the chapter

Response: The title of section 3.3 is revised to capture this concern.

  1. A summary of Chapter 3.3 does not indicate what is written in Chapter. While there is example of critical thinking, for example in solar procurement, but the focus is on the financial issue and economic decisions. The examples presented in the chapter show how much children are aware of environmental problems, which is very impressive.

Response: Summary is revised by putting allocating it in separate a paragraph

  1. The title of chapter 3.4 is Youth Proposed Solutions but lines 464-470, 476-479 and 480-484 related to the title of the chapter

Response: It is unclear what this comment is referring to. The youth proposed solutions as part of our research process in order to start to make changes.

  1. I think what is missing, throughout the article, is what is the research question of the study?

Response: The research question is – What can we learn from rural youth? is the main research question, which is included in the title of the paper. The paper highlights environmental degradation as the answer to this question. The paper has been revised to make this point clearer in the revision.